# Medical Image Understanding with Pre-trained Vision Language Models: A Comprehensive Study

**Ziyuan Qin**[1] *  **Huahui Yi**[1] *  **Qicheng Lao**[2,4] *†  **Kang Li**[1,3,4] †

[1]West China Biomedical Big Data Center, West China Hospital, Sichuan University
[2]School of Artificial Intelligence, BUPT  [3]Sichuan University Pittsburgh Institute  [4]Shanghai AI-Lab
qicheng.lao@bupt.edu.cn    likang@wchscu.cn

## Abstract

The large-scale pre-trained vision language models (VLM) have shown remarkable domain transfer capability on natural images. However, it remains unknown whether this capability can also apply to the medical image domain. This paper thoroughly studies the knowledge transferability of pre-trained VLMs to the medical domain, where we show that well-designed medical prompts are the key to elicit knowledge from pre-trained VLMs. We demonstrate that by prompting with expressive attributes that are shared between domains, the VLM can carry the knowledge across domains and improve its generalization. This mechanism empowers VLMs to recognize novel objects with fewer or without image samples. Furthermore, to avoid the laborious manual designing process, we develop three approaches for automatic generation of medical prompts, which can inject expert-level medical knowledge and image-specific information into the prompts for fine-grained grounding. We conduct extensive experiments on thirteen different medical datasets across various modalities, showing that our well-designed prompts greatly improve the zero-shot performance compared to the default prompts, and our fine-tuned models surpass the supervised models by a significant margin. [1]

## 1 Introduction

There may not exist another domain like medical images that requires high level of expert knowledge, while acquiring expert labeled data is also quite expensive. In fact, limited amount of well-labeled data is one of the factors that deter the medical image domain moves toward the era of large-scale pre-trained models, and transfer learning becomes a natural choice. Nevertheless, as argued in (Niu et al., 2021), the mismatch between domains may compromise the capability of the pre-trained models being transferred from one to another (Raghu et al., 2019). Unfortunately, this mismatch also exists between medical and natural image domains. Therefore, finding a data-efficient approach with superior domain transfer performance is essential for advancing medical image understanding.

Though pre-trained vision-language models (VLMs) have shown much success in domain transfer tasks, it is not known whether the knowledge learned from natural image and text pairs through large pre-trained vision-language models can benefit the understanding of the medical images. As pointed out by (Shen et al., 2022), the large-scale VLMs perform well in recognizing common objects but may not perform well while encountering visual concepts that rarely appeared in their pre-training data. This observation motivates us to discover an even stronger approach to bridge the domain gap. In VL models like GLIP (Li et al., 2022), X-VLM (Zeng et al., 2021), and VinVL (Zhang et al., 2021), prompt learning also plays an essential role in enhancing the model's generalization. Instead of simply aligning the text and image pairs, GLIP aims to ground image regions with the help of text prompts and shows that prompts with expressive attributes can further improve model's performance in domain transfer. We presume that a prompt integrated with expert-level knowledge and image-specific information could vastly help the domain transfer process because one key challenge in

---

*Equal contribution      †Corresponding author
[1]Code and more information could be found at `https://github.com/MembrLab/MIU-VL`

medical image understanding is locating the objects that merely appear in the natural image domain. With the help of well-designed text prompts, the model can be equipped with high-level semantics describing the characteristic of target objects instead of only providing object names.

In this paper, we aim to leverage the powerful pre-trained vision-language models like GLIP with expressive medical prompts to make efficient domain transfers from natural images to medical images for object detection. To this end, we first explore how to manually design effective medical prompts by using attribute injection, and show that such well-designed prompts can significantly improve the domain transfer capability compared to the default category names. Intuitively, some common graphic attributes in text prompts, such as color, texture and shape, are shared across domains, and therefore by including these expressive attributes in the prompts, the VLMs can selectively learn to align visual features through the anchor points set by the prompts rather than aimlessly learning.

Furthermore, to improve the efficiency and avoid the laborious manual designing, we propose several approaches, i.e., masked language model (MLM) driven auto-prompt generation, image specific auto-prompt generation or a hybrid of both, to automatically generate medical prompts that make the VLMs perform on par with the model with manually elaborated prompts. The MLM-driven approach mainly focuses on extracting expert-level knowledge from pretrained language models specialized in the medical domain, whereas the image-specific prompt generation, based on visual question answering (VQA) system, allows the flexibility in designing prompts to include image-specific attribute information, rather than using a single fixed prompt for all images during inference.

We evaluate our approaches on a broad range of existing medical datasets across different modalities including photography, endoscopy, cytology, histopathology and radiology (X-ray, CT, MRI and Ultrasound) image datasets. The models with our well-designed medical prompts exhibit significant superiority compared to those with default prompts in terms of zero-shot and few-shot performance, some even surpassing the supervised model trained with full data. Moreover, our fine-tuned models outperform the traditional supervised baselines by a significant margin across almost all datasets.

## 2 RELATED WORK

**Transfer between natural and medical image domains** Transfer learning is a prevailing strategy for training deep neural networks for domains with limited labeled data, such as the medical domain. Transfer learning has been widely investigated in the medical domain for a while (Peng et al., 2021; Mustafa et al., 2021; Raghu et al., 2019). Zhou et al. (2021) broadly discussed about transfer learning for medical images. Mustafa et al. (2021) argued that transfer from natural to medical images could help if performed at a sufficient scale. Peng et al. (2021) and Raghu et al. (2019) pointed out that large models do not consistently outperform the simple and lightweight models. To the best of our knowledge, there hasn't been any transfer learning work done on medical images with VLMs.

**Vision language models** Recently, VLMs have made breakthroughs in cross-modal tasks and visual recognition problems. Some pre-trained VLMs (Lu et al., 2019; Ilharco et al., 2021) proposed to leverage BERT-like architecture to deal with cross-modal inputs, and Zhang et al. (2020) adopted the contrastive learning paradigm to train a VLM for medical images. Inspired by this line of work, in CLIP (Radford et al., 2021) and ALIGN (Jia et al., 2021), a large amount of image and text pairs have been used to train the VLMs through contrastive learning. Eslami et al. (2021) proposed to leverage large-scale VLMs for medical VQA tasks. While these work focusing on pre-trained VLMs, another line of work focuses on integrating multi-task learning with the vision-language pre-training paradigm (Bao et al., 2021; Yu et al., 2022; Wang et al., 2022). And these models are capable of performing cross-modal tasks, such as image captioning and visual question answering. (Moon et al., 2021) is one of the pioneer works in the medical domain for VLM multi-tasks learning.

**Prompt design** Knowledge-intensive domains, such as the medical domain, usually require training domain-specific language models on expert knowledge augmented corpus to learn proper representations for domain concepts (Gu et al., 2021b; Lee et al., 2020). Moreover, prompting language models in zero-shot or few-shot manner to elicit knowledge has been a commonly adopted approach in recent years (Petroni et al., 2019; Jiang et al., 2020). Except for directly mining knowledge from language models, Shen et al. (2022) designed a pipeline for extracting knowledge from an external source such as WordNet (Miller, 1998). Our proposed auto-prompts generation approaches are also partially inspired by the line of research (Song et al., 2022; Yang et al., 2022). Zhou et al. (2022) pro-

posed to learn to prompt with context optimization. These prompting methods successfully help us to generate knowledge-rich prompts for further VLM prompting in a zero-shot or few-shot manner.

**Object detection and phrase grounding** The R-CNN series are the first to introduce CNNs into the field of object detection and have been a great success (Girshick et al., 2014; Girshick, 2015; Ren et al., 2015). They are two-stage object detector, while single-stage detector networks are more compact, e.g., YOLO (Redmon et al., 2016; Redmon & Farhadi, 2017; 2018), SSD (Liu et al., 2016), RetinaNet (Lin et al., 2017). Later, DyHead (Dai et al., 2021) unifies object detection heads with attentions, improving performance. Recently, GLIP (Li et al., 2022) unifies phrase grounding and object detection tasks, demonstrating exciting domain transfer capability. By making use of the rich knowledge learned from CLIP and text input, ViLD (Gu et al., 2021a) is proposed for open-vocabulary object detection and DenseCLIP (Rao et al., 2022) further improves the performance.

## 3 METHOD

In this work, we mainly explore how to leverage the entailed knowledge in the large vision-language models, such as GLIP (Li et al., 2022), and transfer it to medical domains. Towards this end, we conduct a comprehensive study on a variety of detection tasks in medical domains, where we propose several strategies for better elicitation of medical knowledge from vision-language models pretrained on natural images. We focus on the design and automatic generation of medical prompts that can include expert-level knowledge and image-specific information, which empowers the vision-language models for medical lesion detection in both zero-shot transfer and fine-tuning scenarios.

### 3.1 PRELIMINARIES

Unifying the vision and language pre-training norms has become a prevailing method to enhance the algorithm's performance in many vision-related tasks, showing promising domain transfer capability as well. Following the idea of introducing language supervision into visual recognition problems, GLIP (Li et al., 2022) reformulates object detection as phrase grounding tasks where the model takes both an image input and a text prompt which contains the candidate categories for the target objects. Then both inputs will go through a specific image/text encoder to obtain unaligned representations. During the pre-training stage, GLIP uses a grounding module to align image boxes/regions with corresponding phrases in the text prompt. For example, a prompt can simply be like the following format: Prompt = "$object_1, object_2, object_3...object_M$", where $object_i$ is a class name among the $M$ candidate classes. This alignment/grounding process in GLIP can be formulated as follows:

$$O = Enc_I(\text{Image}), P = Enc_L(\text{Prompt}), S_{ground} = OP^\top, L_{cls} = Loss(S_{ground}; T), \quad (1)$$

where $O \in \mathbb{R}^{N \times d}, P \in \mathbb{R}^{M \times d}$ denote the image and text features respectively, $S_{ground} \in \mathbb{R}^{N \times M}$ represents the cross-modal alignment scores, and $T \in \{0, 1\}^{N \times M}$ is the target matrix. With the aforementioned alignment training by minimizing the loss function, it is not hard to see that the cross-modal inputs have been sufficiently aligned, so one could provide an auxiliary prompt input to guide the image module to locate the corresponding regions more easily. Given that, we believe a well-designed prompt could largely enhance the performance of the pretrained models on the subsequent detection/grounding tasks, especially in an unfamiliar domain like medical images.

### 3.2 MEDICAL PROMPT DESIGN WITH ATTRIBUTE INJECTION

Here, we take the GLIP model as an entry point to explore how to utilize the text prompts and vision-language models entailed knowledge to bridge the gap between the natural and medical image domains smoothly. Similar to previous findings in natural images (Shen et al., 2022; Li et al., 2022), our preliminary experiments also indicate that providing an expressive description in medical prompt can primarily benefit the zero-shot transfer performance of vision language models. More importantly, we find that the injection of shared attributes between natural and medical domains such as shape, color and location would be the most vital in locating the novel categories from the medical domain.

Following this idea, we propose to design medical prompts with a focus on the injection of essential attributes describing the medical objects/lesions of interest. Assuming $M$ categories of target objects

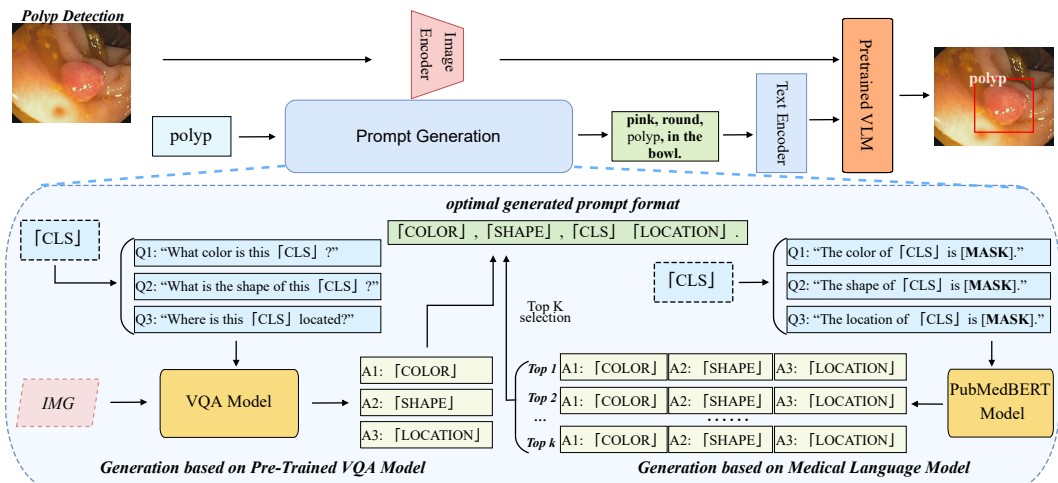

Figure 1: Overview of the proposed approach. The optimal medical prompts can be automatically generated with the help of pre-trained VQA model, medical language model, or a hybrid of both.

associated with $N$ attributes, we can construct the prompt by using the following template:

$$\text{Prompt} = \sum_m \text{Template}(\{v^{Attr_i}\}, object_m), Attr_i \in \{Attr_1, Attr_2, ..., Attr_N\}, \quad (2)$$

where the summation means the concatenation of $M$ categories of objects described by the $N$ chosen attributes. For example, if the attribute set is chosen as {shape, color, location}, then a templated prompt could be '$object$ is $v^{shape}$ shape, in $v^{color}$ color, located on $v^{location}$'. By injecting the specifically engineered attributes, the zero-shot results increase significantly and surpass the results of providing with only the default category name by a large margin. This pattern could be seen in a variety of medical datasets across different modalities, from endoscopy images to histopathological images, demonstrating the effectiveness of well-designed medical prompts with attribute injection.

However, during the process of searching for appropriate prompts, we also find that the current text prompt design has the following limitations: Firstly, manually designing an effective prompt requires expert-level knowledge and a lot of effort; Secondly, in the current vision-language models, the prompts are normally fixed for all samples during inference, i.e., not image-specific, which is not ideal for grounding novel objects that may have varying appearances. For example, malignant skin lesions or diabetic foot wounds often have various irregular shapes and colors.

## 3.3 AUTOMATIC GENERATION OF MEDICAL PROMPTS

To overcome such limitations, in this section, we further investigate how to efficiently generate knowledge-rich and image-specific prompts. Particularly, we discuss about the creative auto-prompt pipelines we proposed for generating expert-level knowledge supported and image-specific prompts.

**Masked Language Model Driven Auto-Prompt Generation** To obtain expert-level knowledge, we utilize medical knowledge enriched (or expert-level) BERT-like pre-trained language models, e.g., the PubMedBERT model (Gu et al., 2021b), for identifying key attributes of a medical concept. The PubMedBERT model is derived from ordinary BERT-like pre-trained NLP models (Petroni et al., 2019) through masked language modeling, but pretrained on the biomedical domain.

Figure 1 (right) illustrates the overall flow of our MLM-driven auto-prompt generation pipeline. We first ask the model which contains medical domain-specific knowledge to predict the masked token in given cloze sentences we design. The template of the cloze sentences is given as: 'The [Attr] of an [Object] is [MASK]', where the 'Attr' and 'Object' tokens are provided and represent the desired attribute name and category name respectively. This operation could be formulated as:

$$v^{Attr} = \arg \max_{\tilde{v}^{Attr} \in V} P_{\text{Expert}}([mask] = \tilde{v}^{Attr} | t_s), \quad (3)$$

where $V$ is the expert knowledge augmented vocabulary, and $t_s$ represent the tokens constituting the cloze sentence template we mentioned above. $P_{\text{Expert}}$ represents the conditional probability of predicting the masked attribute value $\tilde{v}$ for a desired attribute $Attr$ and the target object name $object$.

We take the top-$k$ predicted words for the [MASK] token as our candidate attribute value, because the language model not necessarily always predict the correct word. Then we generate top-$k$ prompts using the template defined in Eq. (2), by repeating the above process for each attribute $Attr_i$ in the attribute set and each object category $object_m$ to be detected. The whole process can be formulated as follows:

$$\text{Prompt}_k = \sum_m \text{Template}(\{v_k^{Attr_i}\}, object_m), Attr_i \in \{Attr_1, Attr_2, ..., Attr_N\}, \quad (4)$$

Where $\{v_k^{Attr_i} | v_k^{Attr_i} \in \underset{\tilde{v}^{Attr_i} \in V}{\text{Top-}k} \{P_{\text{Expert}}([mask] = \tilde{v}^{Attr_i} | t_s)\}\}$ are the top-$k$ attribute values predicted by the masked language model described in Eq. (3), i.e., $MLM(Attr_i, object_m)$.

**Image Specific Auto-Prompt Generation** Although with the above MLM-driven prompt generation approach, we can successfully generate auto-prompts that are supported by expert-level knowledge, the prompts are still not flexible enough to include image-specific attribute information. Therefore, in this section, we further propose an image specific auto-prompt generation approach by adopting pre-trained visual question answering (VQA) models, e.g., the OFA model (Wang et al., 2022). As demonstrated in Figure 1 (left), we ask the VQA models multiple questions related to the desired attributes iteratively. For example, we can ask the model: "What color is this wound?". We expect to receive a proper answer from the VQA model and take that answer as the value for the related attribute. Unlike the MLM-driven approach, we won't ask for top-$k$ answers due to the computation time constraint. This process has to be applied to each image input to generate image-specific prompts, which means the corresponding prompt for each image can vary. Given an image input $x$, the corresponding prompt could be formulated as follows:

$$\text{Prompt}_x = \sum_m \text{Template}(\{VQA(x, Q_{Attr_1}), ..., VQA(x, Q_{Attr_N})\}, object_m), \quad (5)$$

where $VQA(x, Q_{Attr_i})$ represents the attribute value obtained from the VQA model with image input $x$ and question $Q_{Attr_i}$ for the $i$-th attribute, and Template$(\cdot)$ is the same as defined in Eq. (2).

We believe that the domain transfer performance would be improved if we inject both expert-level knowledge and image-specific information into the prompts. However, our preliminary results obtained from the VQA prompts suggest that certain attribute (e.g., location) may not be appropriately answered by the pre-trained VQA models. We speculate that the wrong locations given by the VQA models can be explained by the fact that most of the medical images are taken in a quite different environment compared to the natural images, and therefore expecting the VQA model pre-trained on natural images to recognize what organ or which part of the human body is in the image could be challenging. In this regard, we choose to combine the two above approaches, namely the MLM-driven approach and the VQA based approach for different attributes. For example, we can use the VQA models to provide the object intrinsic attributes such as shape, color and texture, while for the location attribute, we obtain it from the language model approach. The intuition behind such a combination is that we think the shape, color and texture of an object are much easier to tell from the image and belong to image-specific characteristics, whereas the location of an object can be ambiguous between the relative location of the object in the image versus the location of which part of human body for medical image grounding. We call the prompts generated by this hybrid approach the 'hybrid prompts', while the ones generated by purely VQA based models are the 'VQA prompts'. In this case, the prompt template in Eq. (5) for 'hybrid prompts' can be updated to:

$$\text{Prompt}_x = \sum_m \text{Template}(\{VQA(x, Q_{Attr_1}), ..., MLM(Location, object_m)\}, object_m), \quad (6)$$

where $MLM(Location, object_m)$ represents the location attribute predicted by the MLM model in Eq. (3), and $VQA(x, Q_{Attr_i})$ represents the VQA model output for other object intrinsic attributes.

## 4 EXPERIMENTS

### 4.1 SETUP

**Datasets.** For a comprehensive study, we collect 13 public medical image datasets across various different modalities including: Photography image datasets for skin lesions ISIC 2016 (Gutman

Table 1: Dataset overview (13 datasets in total).

| | Photography images | | Endoscopy images | Microscopy images | | Radiology images | | | |
|---|---|---|---|---|---|---|---|---|---|
| | | | | Cytology | Histopathology | X ray | CT | MRI | Ultrasound |
| Dataset | ISIC 2016 | DFUC 2020 | Ployp Benchmark (×5)* | BCCD | CPM-17 | TBX11K | Luna16 | ADNI | TN3k |

\* includes CVC-300, CVC-ClinicDB, CVC-ColonDB, Kvasir, and ETIS

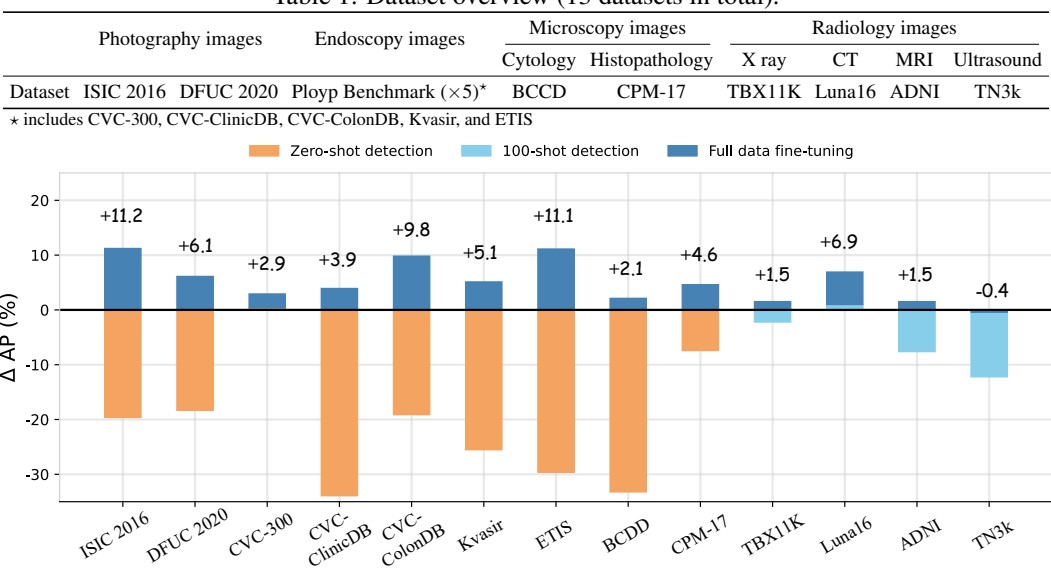

Figure 2: Comparisons with a fully supervised baseline (the horizontal line). The y-axis shows $\Delta$AP compared to the supervised baseline. For non-radiology datasets, we exhibit zero-shot and full data results; we show 100-shot and full data results for the radiology datasets (from TBX11K to TN3k).

et al., 2016) and diabetic foot ulcer DFUC 2020 (Cassidy et al., 2021); Endoscopy image datasets for polyp detection CVC-300 (Vázquez et al., 2017), CVC-ClinicDB (Bernal et al., 2015), CVC-ColonDB (Tajbakhsh et al., 2015), Kvasir (Jha et al., 2020) and ETIS (Silva et al., 2014); Cytology image dataset BCCD; Histopathology image dataset CPM-17 (Vu et al., 2019); and Radiology image datasets TBX11k (Liu et al., 2020), Luna16 (Setio et al., 2017), ADNI (Boccardi et al., 2015) and TN3k (Gong et al., 2021) for X ray, CT, MRI and ultrasound, respectively. Table 1 summarizes the datasets and more details on the data split and prepossessing are included in the Appendix.

**Implementation details.** For our experiments, we use the GLIP-T (C) variant (Li et al., 2022) as our base pre-trained model and follow their hyper-parameter choices when transferring to medical images. We train our models using Adam optimizer with base learning rate of $1 \times 10^{-4}$ ($1 \times 10^{-5}$ for the BERT text encoder), and the weight decay is set to 0.05. We freeze the bottom two layers of the image encoder and decay the learning rate by 0.1 when the validation performance plateaus. For the language-driven automatic prompt generation, we use the PubmedBert-base-uncased variant (Gu et al., 2021b) to fill the cloze sentences. Moreover, we use the OFA-base variant (Wang et al., 2022) and its VQA module to generate the attribute values automatically. For the comparison experiments, we use the supervised models Faster RCNN (Ren et al., 2015), RetinaNet (Lin et al., 2017) and DyHead (Dai et al., 2021) provided by the MMDetection framework (Chen et al., 2019).

## 4.2 Transfer to Established Medical Benchmarks

This section demonstrates that the GLIP model, with the aid of well-designed language prompts, can directly or indirectly transfer to the medical domain with competitive performance. For convenience, we split the medical datasets into two major categories: non-radiology and radiology datasets. In the following we first give an overview of our fine-tuned models surpassing the supervised baseline. Then, we illustrate the results of the proposed approach on non-radiology datasets, focusing on the zero-shot scenario. Finally, we discuss the fine-tuning results on the radiology datasets.

**Transfer performance surpassing supervised methods** To prove that text prompts are effective for cross-domain transfer, we conduct extensive experiments under both zero-shot domain transfer and supervised transfer (fine-tuning) settings. We include a series of supervised baselines: Faster-RCNN, RetinaNet, and DyHead-L for comparisons. As illustrated in Figure 2, our full data fine-tuned models with well-designed medical prompts (*dark blue*) surpass the supervised baseline (e.g., DyHead-L in the figure) by a large margin across all datasets. Moreover, even zero-shot (*brown*) or 100-shot (*sky blue*) results on some datasets, e.g., CVC-300 and Luna-16, can rival the full data

Table 2: Our approaches v.s. supervised models on non-radiology datasets (AP%).

| | Method | Backbone | ISIC 2016 | DFUC 2022 | Polyp (×5) | BCCD | CPM-17 | Avg. |
|---|---|---|---|---|---|---|---|---|
| Full Data | Faster RCNN | RN50 | 50.3 | 42.3 | 56.6 | 56.9 | 39.8 | 49.2 |
| | RetinaNet | RN50 | 54.0 | 43.1 | 58.8 | 56.7 | 35.7 | 49.7 |
| | DyHead | Swin-T | 52.9 | 44.2 | 62.9 | 60.1 | 38.8 | 51.8 |
| | GLIP-T(default cls) | Swin-T | 62.4 | 50.3 | 68.1 | 62.5 | 43.9 | 57.4 |
| | Ours (Manual) | Swin-T | 64.1 | 50.3 | 69.4 | 62.2 | 43.4 | 57.9 |
| | Ours (Auto) | Swin-T | 61.6 | 50.1 | 68.8 | 63.1 | 44.2 | 57.6 |
| 100-Shot | Faster RCNN | RN50 | 44.6 | 27.0 | 44.9 | 38.6 | – | 38.8 |
| | RetinaNet | RN50 | 41.7 | 28.4 | 41.7 | 54.3 | – | 41.5 |
| | DyHead | Swin-T | 42.5 | 27.8 | 42.5 | 40.5 | – | 38.3 |
| | GLIP-T(default cls) | Swin-T | 55.9 | 41.4 | 57.6 | 59.8 | – | 53.7 |
| | Ours (Manual) | Swin-T | 58.0 | 43.7 | 60.8 | 60.1 | – | 55.7 |
| | Ours (Auto) | Swin-T | 58.8 | 42.4 | 60.8 | 60.2 | – | 55.6 |
| Zero-Shot | GLIP-T(default cls) | Swin-T | 20.1 | 0.1 | 4.1 | 0.7 | 7.6 | 6.5 |
| | GLIP-L(default cls) | Swin-L | 20.4 | 3.6 | 11.9 | 10.4 | 11.6 | 11.6 |
| | Ours (with MLM) | Swin-T | 25.1 | 24.8 | 38.4 | 24.1 | 20.3 | 26.5 |
| | Ours (with VQA) | Swin-T | 23.5 | 12.9 | 27.1 | 14.3 | 26.2 | 20.8 |
| | Ours (with Hybrid) | Swin-T | 24.5 | 22.5 | 35.1 | 14.3 | 24.8 | 24.2 |
| | Ours (Manual) | Swin-T | 33.3 | 25.9 | 41.3 | 26.9 | 31.4 | 31.8 |

Figure 3: Left: Attribution injection in the prompts improves the detection performance; Right: Data efficiency comparison between vision language models and classical detection models (Kvasir).

fine-tuned supervised models. The quantitative numbers are respectively shown in Table 2 for non-radiology datasets, Table 3 for polyp datasets, and Table 5 for radiology datasets. This is also supported by Figure 3 (right) where the VLMs significantly outperform the classical detection models with fully supervised learning, especially in few-shot settings.

**Superior zero-shot transfer performance compared to the baseline**  Here, we provide strong evidence to show our approaches can empower the pre-trained VLM with remarkable zero-shot capability in the medical domain. As shown in Table 2 and Table 3, the prompts generated by our approaches tremendously improve the zero-shot performance of the GLIP models compared to the default ones. For example, on the polyp benchmarks, the out-of-box GLIP-T model only achieves an average AP of 4.1%, while the same model with our manually designed prompts reaches 41.3%. And this massive gain is not an exception. In addition, the models with well-designed prompts can reach an overall performance on par with the 100-shot fine-tuned baseline models on the polyp benchmarks (Table 3), and sometimes even rival the supervised baseline models trained with full-size data, e.g., on the CVC-300 dataset (69.9% AP for zero-shot *v.s.* 59.4% for Faster RCNN).

**The effectiveness of attribution injection and auto-prompts**  In section 3.2, we discussed that adding attributes could make the models perform better in zero-shot tasks. Here, we demonstrate in Figure 3 (left) an overall pattern of the effect of attribute injection on performance under the zero-shot setting. As shown in the figure, the overall performance increases as more attributes are

Table 3: Our approaches v.s. supervised models on polyp benchmark datasets (AP%).

|  | Method | Backbone | CVC-300 | CVC-ClinicDB | CVC-ColonDB | Kvasir | ETIS | Avg. |
|---|---|---|---|---|---|---|---|---|
| Full Data | Faster RCNN | RN50 | 59.4 | 71.6 | 44.1 | 63.4 | 44.5 | 56.6 |
| | RetinaNet | RN50 | 61.6 | 71.9 | 49.8 | 64.1 | 46.6 | 58.8 |
| | DyHead | Swin-T | 69.5 | 73.5 | 51.4 | 68.6 | 51.3 | 62.9 |
| | GLIP-T | Swin-T | **75.0** | 71.9 | 60.9 | 69.8 | **62.8** | 68.1 |
| | Ours (Manual) | Swin-T | 72.4 | 77.4 | **61.2** | **73.7** | 62.4 | **69.4** |
| | Ours (Auto) | Swin-T | 72.0 | **78.7** | 61.0 | 70.9 | 61.6 | 68.8 |
| 100-Shot | Faster RCNN | RN50 | 53.6 | 41.2 | 27.2 | 43.9 | 26.9 | 38.6 |
| | RetinaNet | RN50 | 54.1 | 46.8 | 30.6 | 47.5 | 29.4 | 41.7 |
| | DyHead | Swin-T | 54.0 | 45.0 | 27.6 | 45.4 | 30.4 | 40.5 |
| | GLIP-T | Swin-T | 69.6 | 59.4 | 52.3 | 63.0 | 43.6 | 57.6 |
| | Ours (Manual) | Swin-T | 70.2 | **61.6** | 53.6 | 66.8 | **51.8** | **60.8** |
| | Ours (Auto) | Swin-T | **71.3** | 60.4 | **55.4** | 67.1 | 49.6 | **60.8** |
| Zero-Shot | GLIP-T | Swin-T | 6.1 | 4.1 | 3.2 | 7.2 | 0.1 | 4.1 |
| | GLIP-L | Swin-L | 10.3 | 9.9 | 7.4 | 24.9 | 7.1 | 11.9 |
| | Ours (with MLM) | Swin-T | 64.1 | 38.3 | 27.4 | **45.0** | 17.0 | 38.4 |
| | Ours (with VQA) | Swin-T | 54.4 | 22.8 | 16.1 | 28.1 | 14.2 | 27.1 |
| | Ours (with Hybrid) | Swin-T | 63.2 | 31.4 | 20.0 | 37.2 | **23.6** | 35.1 |
| | Ours (Manual) | Swin-T | **69.9** | **39.6** | **32.3** | 43.1 | 21.7 | **41.3** |

Table 4: Examples of prompts for BCDD (zero-shot performance on the validation and test set)

|  | Prompt | AP | AP50 |
|---|---|---|---|
| initial | platelet. red blood cell. white blood cell | 0.4 | 0.9 |
| medical concepts | thrombocyte. erythrocyte. leukocyte | 0.1 | 0.1 |
| | blood platelet. red blood corpuscle. white blood corpuscle | 3.1 | 7.0 |
| | thrombocyte, blood platelet. erythrocyte, red blood corpuscle. leukocyte, white blood corpuscle | 6.8 | 15.5 |
| | **thrombocyte or** blood platelet. **erythrocyte or** red blood **corpuscle**. **leukocyte or** white blood **corpuscle** | 8.6 | 17.9 |
| + location | platelet **in blood**. red blood cell **in blood**. white blood cell **in blood** | 6.9 | 14.4 |
| + shape | **small** platelet. **rounded** red blood cell. **irregular** white blood cell | 7.7 | 14.9 |
| + color | colorless platelet. freshcolor red blood cell. blue white blood cell | 18.3 | 32.3 |
| | colorless platelet. freshcolor red blood cell. purple white blood cell | 17.8 | 32.9 |
| | **colorless** platelet. **freshcolor** red blood cell. **purple or blue** white blood cell | 24.9 | 43.8 |
| combinations | small, colorless platelet. rounded, freshcolor red blood cell. irregular , purple or blue white blood cell | 26.6 | 47.1 |
| | small, colorless blood platelet. rounded, freshcolor erythrocyte. irregular, purple or blue leukocyte | 26.4 | 45.3 |
| | **small**, **colorless** platelet. **rounded**, **freshcolor** red blood **corpuscle**. **irregular**, **purple or blue** white blood **corpuscle** | **27.1** | **47.6** |

integrated into the prompts. This is also illustrated in Table 4 on the BCCD dataset, where various attributes and their combinations are shown to improve the results. As this process is rather tedious and time consuming, we need qualified automatic approaches to accelerate the generation process to scale up without sacrificing too much performance. Fortunately, the models with our proposed auto-prompts, especially with the hybrid and MLM-driven approaches, show comparable results to those with manually created prompts and surpass those with default prompts by a landslide. For example, the MLM-driven approach achieves an AP of 24.8% for zero-shot on the DFUC2022 dataset, while the GLIP-T baseline with default prompts only gives 0.1% for the zero-shot performance (Table 2). Figure 4 shows an example of the auto-prompt generation with the hybrid approach.

**Fine-tuning on radiology datasets** We finally evaluate the fine-tuned models on the radiology datasets under different few-shot settings, i.e., 1-shot, 10-shot, 100-shot, as well as full-data fine-tuning. The results are presented in Table 5. As we can see from the numbers, the overall performance of our fine-tuned models is much better than that of the supervised baselines, consistent with the findings in non-radiology medical images. This property reveals itself extremely in the 1-shot experiments. The average AP across all datasets of our models reaches 5.7 %AP, while other baselines give 0% AP. As the training data increases from zero-shot to full-size, the performance gap gets narrower. According to this pattern, we conclude that the pre-trained VLMs like GLIP is more data efficient than the traditional supervised baselines. *Given the medical image data's scarcity, we believe the data efficient property of VLMs would benefit many medical scenarios.*

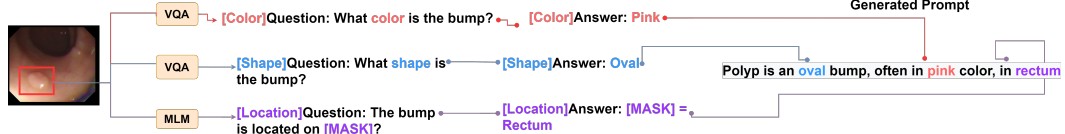

Figure 4: Auto-prompt generation show case.

Table 5: Radiology images requires fine-tuning. Prompts: pulmonary tuberculosis (TBX11K); lung nodule (Luna16); hippocampus (ADNI); thyroid nodule (TN3k).

| | Method | Backbone | TBX11K | | Luna16 | | ADNI | | TN3k | | Avg. | |
|---|---|---|---|---|---|---|---|---|---|---|---|---|
| | | | AP | AP50 | AP | AP50 | AP | AP50 | AP | AP50 | AP | AP50 |
| 1-Shot | Faster RCNN | RN50 | 0.0 | 0.3 | 0.0 | 0.0 | 0.0 | 0.0 | 0.0 | 0.3 | 0.0 | 0.2 |
| | RetinaNet | RN50 | 0.0 | 0.0 | 0.0 | 0.0 | 0.0 | 0.0 | 0.0 | 0.0 | 0.0 | 0.0 |
| | DyHead | Swin-T | 0.0 | 0.0 | 0.0 | 0.0 | 0.0 | 0.0 | 0.0 | 0.0 | 0.0 | 0.0 |
| | Ours | Swin-T | **6.0** | **19.8** | **2.5** | **6.4** | **1.7** | **5.5** | **12.7** | **24.2** | **5.7** | **14.0** |
| 10-Shot | Faster RCNN | RN50 | 3.2 | 13.4 | 0.0 | 0.0 | 0.1 | 0.3 | 1.0 | 4.4 | 1.1 | 4.3 |
| | RetinaNet | RN50 | 4.5 | 16.3 | 0.0 | 0.0 | 0.6 | 2.7 | 1.0 | 4.1 | 1.5 | 5.8 |
| | DyHead | Swin-T | 1.9 | 6.5 | 0.0 | 0.1 | 0.5 | 2.1 | 1.4 | 5.4 | 1.0 | 3.5 |
| | Ours | Swin-T | **13.0** | **37.6** | **12.3** | **32.8** | **15.1** | **44.9** | **26.1** | **49.6** | **16.6** | **41.2** |
| 100-Shot | Faster RCNN | RN50 | 28.6 | 71.5 | 15.3 | 45.9 | 29.5 | 73.4 | 29.1 | 65.2 | 25.6 | 64.0 |
| | RetinaNet | RN50 | 29.8 | **73.3** | 4.4 | 16.9 | 27.5 | 72.9 | 33.6 | 69.5 | 23.8 | 58.1 |
| | DyHead | Swin-T | 28.5 | 70.4 | 23.2 | 61.8 | 28.9 | 71.1 | 33.5 | 71.5 | 28.5 | 68.7 |
| | Ours | Swin-T | **33.5** | 72.6 | **34.1** | **78.1** | **39.5** | **77.0** | **48.5** | **79.2** | **38.9** | **76.7** |
| Full Data | Faster RCNN | RN50 | 33.9 | 73.9 | 32.0 | 69.5 | 46.5 | 80.8 | 54.1 | 84.9 | 41.6 | 77.3 |
| | RetinaNet | RN50 | 37.0 | 77.9 | 32.3 | 76.1 | **48.9** | **82.8** | 56.8 | 88.0 | 43.8 | 81.2 |
| | DyHead | Swin-T | 35.7 | 74.4 | 33.1 | 82.3 | 47.1 | 81.2 | **60.7** | **90.9** | 44.2 | 82.0 |
| | Ours | Swin-T | **37.2** | **78.5** | **40.0** | **84.7** | 48.6 | **82.8** | 60.3 | 90.7 | **46.5** | **84.2** |

Table 6: Ablation on the input size and freeze layers in image and text encoders (Luna16).

| Size | Image Encoder | | | | Text Encoder | AP | AP50 |
|---|---|---|---|---|---|---|---|
| | layer0 | layer1 | layer2 | layer3 | | | |
| $512 \times 512$ | | ✓ | | | | 34.2 | 73.0 |
| $800 \times 800$ (default) | | | | | | 38.0 | 78.6 |
| | ✓ | ✓ | | | | 40.0 | **84.7** |
| | ✓ | ✓ | ✓ | ✓ | | 37.2 | 78.8 |
| | ✓ | ✓ | | | ✓ | **40.1** | 79.1 |
| | ✓ | ✓ | ✓ | ✓ | ✓ | 31.9 | 78.8 |

**Ablation studies** Table 6 presents the ablation studies on the image input size and freeze layers in the image and text encoders on the Luna16 lung CT dataset. As shown in the table, our default choice of using input size at $800 \times 800$ (input size used in pre-training) is much better than using the dataset specific size (i.e., $512 \times 512$ for Luna16). For the freeze layers, we choose to freeze the bottom two layers as in GLIP (Li et al., 2022), and we find complete freeze of the visual backbone or no freeze at all are not the best choice for VLM domain transfer. With the visual backbone set by default, freezing the linguistic backbone has little impact on the model performance.

## 5 CONCLUSION

This paper comprehensively studies how to leverage the large-scale vision language models pre-trained on natural images to medical images. We present that well-designed medical prompts containing domain-specific knowledge is the key to bridging the gap between domains. Therefore, we propose several approaches to generate medical prompts in either manual or automatic manners. While the manual approach tremendously improves the zero-shot performance compared to the default prompts with object names, the automatic approaches allow us to generate expert knowledge augmented and image-specific prompts on a large scale. Extensive experiments are conducted on thirteen different medical datasets across various modalities, showing the the prompts generated by our approaches can improve the transfer performance, and our fine-tuned models surpass the supervised baselines by a large margin. This superior domain transfer performance also prompts us to explore more data-efficient vision-language algorithms to benefit medical image understanding.

ACKNOWLEDGMENTS

We would like to thank the GLIP team and the OFA team for providing excellent code and models to the open source community. We also thank the individuals and organisations who have contributed open source medical imaging datasets. This study was supported by National Key Research and Development Program of China (2020YFB1711500, 2020YFB1711503), and the 1·3·5 project for disciplines of excellence, West China Hospital, Sichuan University (ZYYC21004).

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

## A  Dataset Introduction

In this section, we present the composition details of every dataset we collected. As we mention before, we divide the datasets into two major categories: the non-radiology and radiology datasets.

For non-radiology images, we have photograph images, endoscopy images, cytology images, and histopathology images. Photography images is composed of the ISIC 2016 (Gutman et al., 2016) and DFUC 2020 (Cassidy et al., 2021) dataset. The ISIC-16 dataset consists of 1,279 images with 1,282 bboxes for benign skin lesions and melanoma detection, divided into 720/180/379 images for training, validation, and testing. The DFUC2020 dataset is the largest diabetic foot ulcer detection dataset for now, including 2,000 images, 2,496 bboxes, and 1 class; and those images are divided into 1,280/320/400 images for training, validation, and testing.

For endoscopy images, we use a benchmark composed of a series of datasets for polyp region detection from PraNet (Fan et al., 2020), which includes the following datasets: CVC-300, CVC-ClinicDB, CVC-ColonDB, Kvasir, and ETIS. There are 2,248 images and 2,374 bboxes in total. The complete training and validation images for the entire benchmark are 1160 and 290, respectively. And the number of test set images for CVC-300, CVC-ClinicDB, CVC-ColonDB, Kvasir, and ETIS datasets are 60, 62, 380, 100, and 196 respectively.

For microscopy images, we have two datasets: cytology dataset BCCD and histopathology dataset CPM-17 (Vu et al., 2019). The BCCD dataset is designed for blood cell detection tasks, including three classes: white blood cells, red blood cells, and platelets. There are 874 images with 11,789 bboxes for the entire BCCD dataset. Furthermore, the dataset is split into training, validation, and test sets with 765, 73, and 36 images, respectively. CPM-17 is a cell nuclear detection dataset that contains only one class and consists of 64 images with 7,506 bbox labels. The dataset is divided into 25/7/32 images for training, validation, and testing.

For some datasets, such as the ISIC 2016, Ployp Benchmark, Luna16, ADNI, and TN3k datasets, the bbox labels are obtained from the mask labels of the original dataset, while the labels of dataset CPM-17 are obtained from the instance segmentation labels. For other datasets, we simply use the original bbox labels.

We select a representative dataset for each of the four different modalities of radiology images, includes X-ray dataset TBX11K (Liu et al., 2020), CT dataset Luna16 (Setio et al., 2017), MRI dataset ADNI (Boccardi et al., 2015) and ultrasound dataset TN3k (Gong et al., 2021). The TBX11K dataset is used for tuberculosis detection in the lung, including 799 images and 1,211 bbox labels. Moreover, this dataset is divided into 479/120/200 images for training, validation, and testing sets, respectively. The ADNI dataset is designed for the hippocampus gland detection task, which contains 1186 images and 1186 bboxes. The training, validation, and test sets consist of 759, 190, and 237 images, respectively. The Luna16 is a lung nodule detection dataset, including 3,997 images and 7,545 bbox labels. There are 2,590, 589, and 818 images for training, validation, and test sets. The TN3k dataset is a large thyroid nodule detection in ultrasound images containing 3,493 images and 3811 bboxes. Moreover, the datasets' training, validation, and testing sets consist of 2,303, 576, and 614 images, respectively.

## B  Prompt Generation Implementation Details

To automatically generate prompts for unseen concepts, we design various pipelines for obtaining external knowledge from different sources. For the Language Model (LM) based method, as demonstrated in the methodology section, we use PubmedBERT as our knowledge source and use a template to elicit the attributes' knowledge. For example, if we want to obtain the potential color, shape, and location of polyps from the LM, we will make a template such as "The color of polyps is [Masked]" and ask the language model to predict the "[Masked]" token. Since the BERT-like language model is all pre-trained with the masked token prediction task, the above method can elicit the most likely word for the masked token. Therefore, the LM will give us a probability distribution over all tokens in its vocabulary, and we can take the tokens with top-3 probability as our answers. Furthermore, we use the predicted color as the attribute value for the unseen object to make up our prompts. So, in a nutshell, we first make up a template with the masked token for each attribute, then we use the LM model to do the masked token prediction task to obtain the values. After we

collect all the attribute values we need, we then fill these values into our pre-defined prompt template. For example, suppose we receive the words such as "pink", "round", and "rectum" for the color, shape, and location attributes of polyps. In that case, we fill the template "Polyp is a [color] and [shape] bump in [location]" with the corresponding words to obtain our prompt: "Polyp is a pink and round bump in rectum." One can directly use this sentence as the final version to be fed to the GLIP model. However, because of the implementation detail of the GLIP model which we will not elaborate here, it is better to rearrange the sentence above to a format of a composition of phrases, such as "pink, round, bump, in rectum". The words before the word "bump" will be treated as a prefix, while the words after the word "bump" will be treated as suffix by the GLIP model. For further detail of this arrangement, please refer to the code of the GLIP model.

The workflow of the image-specific VQA method is quite similar to the pipeline above, except we change the knowledge source from the LM model to the VQA model. And for the VQA model, we don't ask the model to predict a masked token; we let it answer our pre-defined question for each attribute and collect the answers.

The hybrid approach is simply a combination of the LM-based and VQA-based methods. We use the VQA model to get the shape and color of the unseen objects since these attributes can vary from image to image. We then use the LM model to predict the possible location of the unseen concepts. After all, we combined the attribute values received from both methods to fill into the prompt template.

## C  THUMB RULES OF DESIGNING PROMPTS

We summarize an empirical rule for manually generating prompts because these rules provide helpful insight into the essence of prompt designing. The first rule is that the more common the target object is, the less expressive characteristics are needed. We argue that since the VLMs have seen the general target objects in their pre-training stage, simply providing the common object names would be enough to activate the learned knowledge. In the DFUC2020 dataset for example, we observe that only providing the location attributes would be enough for the prompt to help the model to achieve the best performance. The target object here is a wound, a fairly common concept not only seen in medical images. In the Polyp benchmark datasets however, we observe the exact opposite case. The target object is a polyp, a rather less general concept that arguably only appears in the medical domain. In this case, we tried including many graphic attributes, such as color, shape, and texture, to obtain an ideal performance.

## D  EXTRA EXAMPLES OF MANUAL PROMPT DESIGN

In this section, we would like to provide extra examples of manually designed prompts and the corresponding results, on both non-radiology and radiology datasets. The following tables show that the color, shape, and location attributes can significantly improve the results.

Table 7: Examples of prompts for CVC-300 (zero-shot performance on the validation and test set)

|  | Prompt | AP | AP50 |
|---|---|---|---|
| initial | polyp | 1.1 | 3.0 |
| medical concepts | polyp is an **abnormal growth** on the surface | 7.0 | 13.1 |
| + shape + general | polyp is a **bump**. | 11.6 | 19.0 |
|  | polyp is an **oval bump**. | 13.9 | 22.0 |
| + shape + color | polyp is an oval bump, often in **flesh pink** color. | 15.4 | 27.6 |
|  | polyp is an oval bump, often in **pink** color. | 28.6 | 50.7 |
| + shape + color + location | In **colon** polyp is an oval bump, often in pink color | 27.8 | 47.0 |
|  | In **rectum** polyp is an oval bump, often in pink color | **43.4** | **69.4** |

Table 8: Examples of prompts for TN3k (zero-shot performance on the validation and test set)

| | Prompt | AP | AP50 |
|---|---|---|---|
| initial | thyroid nodule | 1.9 | 4.2 |
| wikipedia | thyroid nodules are nodule which commonly arise within an otherwise normal thyroid gland | 5.6 | 10.8 |
| + description | **irregular** thyroid tumor. | 4.8 | 11.2 |
| | **salient** thyroid tumor. | 5.5 | 11.1 |
| + domain | thyroid tumor in **medical imaging**. | 11.2 | 20.3 |
| | thyroid tumor in **medical ultrasound imaging**. | 11.3 | 20.9 |
| + description + domain | **salient** thyroid tumor in **medical ultrasound imaging** | **12.2** | **21.4** |

# E STANDARD DEVIATION AND ERROR-BAR FOR FEW-SHOT RESULTS

In this section, we demonstrate the standard deviation numbers and error-bar for our fine-tuning results. We use 3 different random seeds for our few-shot learning experiments to test whether our fine-tuning results are consistent across different random settings. The relative small standard deviation indicates that our method is not sensitive to the randomness.

Table 9: The Mean and Standard Deviation results of Table 2 (AP%).

| | Method | ISIC 2016 | DFUC 2022 | Polyp | BCCD | Avg. |
|---|---|---|---|---|---|---|
| | GLIP-T | $55.9_{\pm 1.67}$ | $41.4_{\pm 0.37}$ | $57.6_{\pm 1.10}$ | $59.8_{\pm 1.15}$ | 53.7 |
| 100-Shot | Ours (Manual) | $58.0_{\pm 1.08}$ | $43.7_{\pm 1.17}$ | $60.8_{\pm 0.64}$ | $60.1_{\pm 0.95}$ | **55.7** |
| | Ours (Auto) | $58.8_{\pm 1.93}$ | $42.4_{\pm 1.06}$ | $60.8_{\pm 1.24}$ | $60.2_{\pm 0.36}$ | 55.6 |

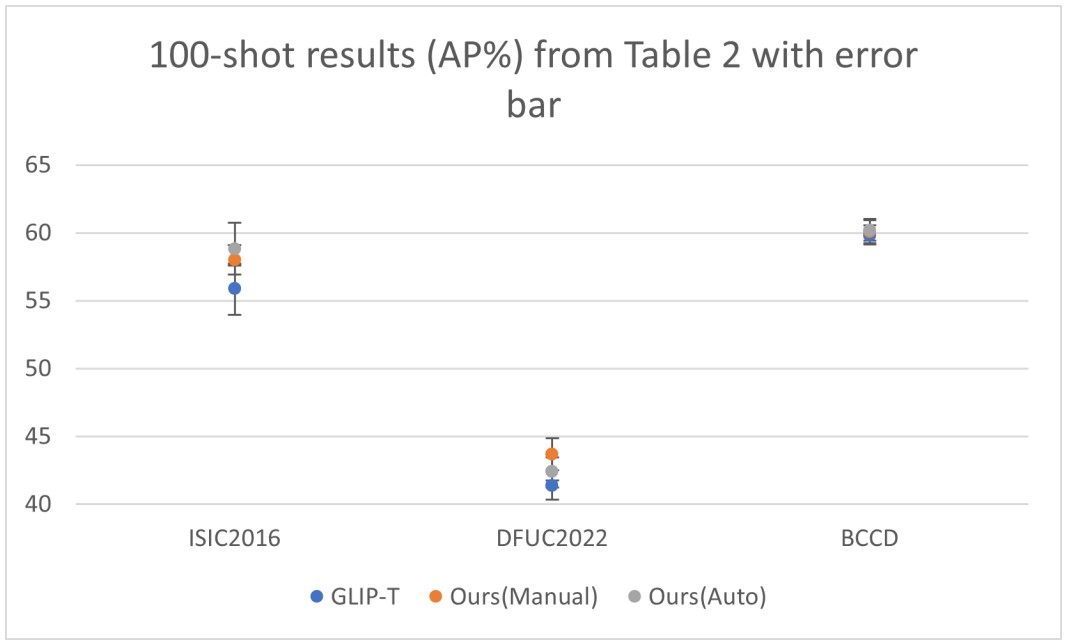

Figure 5: 100-shot results (AP%) from Table 2 with error bar

Table 10: The deviation of the mean results of Table 3 (AP%).

| | Method | CVC-300 | CVC-ClinicDB | CVC-ColonDB | Kvasir | ETIS | Avg. |
|---|---|---|---|---|---|---|---|
| 100-Shot | GLIP-T | $69.6_{\pm 2.42}$ | $59.4_{\pm 1.63}$ | $52.3_{\pm 0.38}$ | $63.0_{\pm 1.44}$ | $43.6_{\pm 3.69}$ | 57.6 |
| | Ours (Manual) | $70.2_{\pm 1.96}$ | $61.6_{\pm 0.88}$ | $53.6_{\pm 2.61}$ | $66.8_{\pm 2.63}$ | $51.8_{\pm 1.94}$ | **60.8** |
| | Ours (Auto) | $71.3_{\pm 0.93}$ | $60.4_{\pm 1.25}$ | $55.4_{\pm 2.36}$ | $67.1_{\pm 1.31}$ | $49.6_{\pm 4.31}$ | **60.8** |

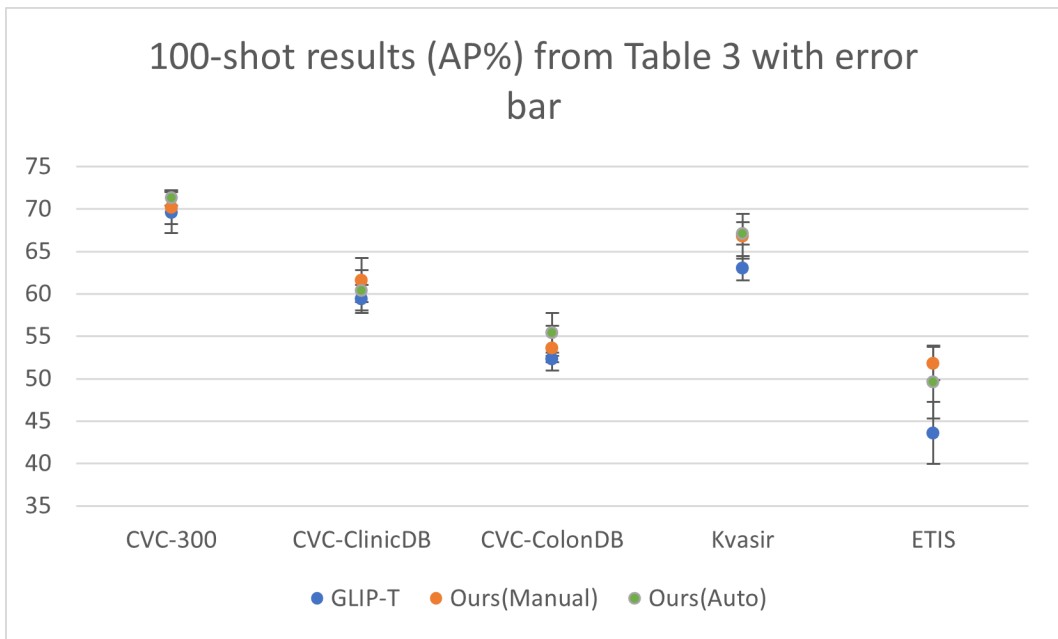

Figure 6: 100-shot results (AP%) from Table 3 with error bar

Table 11: Radiology fine-tuning results with standard deviation using our approaches.

| | TBX11K | | Luna16 | | ADNI | | TN3k | | Avg. | |
|---|---|---|---|---|---|---|---|---|---|---|
| | AP | AP50 | AP | AP50 | AP | AP50 | AP | AP50 | AP | AP50 |
| 1-Shot | $6.0_{\pm 1.10}$ | $19.8_{\pm 4.66}$ | $2.5_{\pm 2.06}$ | $6.4_{\pm 4.67}$ | $1.7_{\pm 0.95}$ | $5.5_{\pm 3.48}$ | $12.7_{\pm 5.5}$ | $24.2_{\pm 8.18}$ | 5.7 | 14.0 |
| 10-Shot | $13.0_{\pm 3.49}$ | $37.6_{\pm 10.10}$ | $12.3_{\pm 1.14}$ | $32.8_{\pm 3.09}$ | $15.1_{\pm 0.59}$ | $44.9_{\pm 3.15}$ | $26.1_{\pm 1.60}$ | $49.6_{\pm 3.77}$ | 16.6 | 41.2 |
| 100-Shot | $33.5_{\pm 1.35}$ | $72.6_{\pm 2.86}$ | $34.1_{\pm 1.81}$ | $78.1_{\pm 2.78}$ | $39.5_{\pm 0.93}$ | $77.0_{\pm 0.70}$ | $48.5_{\pm 0.66}$ | $79.2_{\pm 1.52}$ | **38.9** | **76.7** |

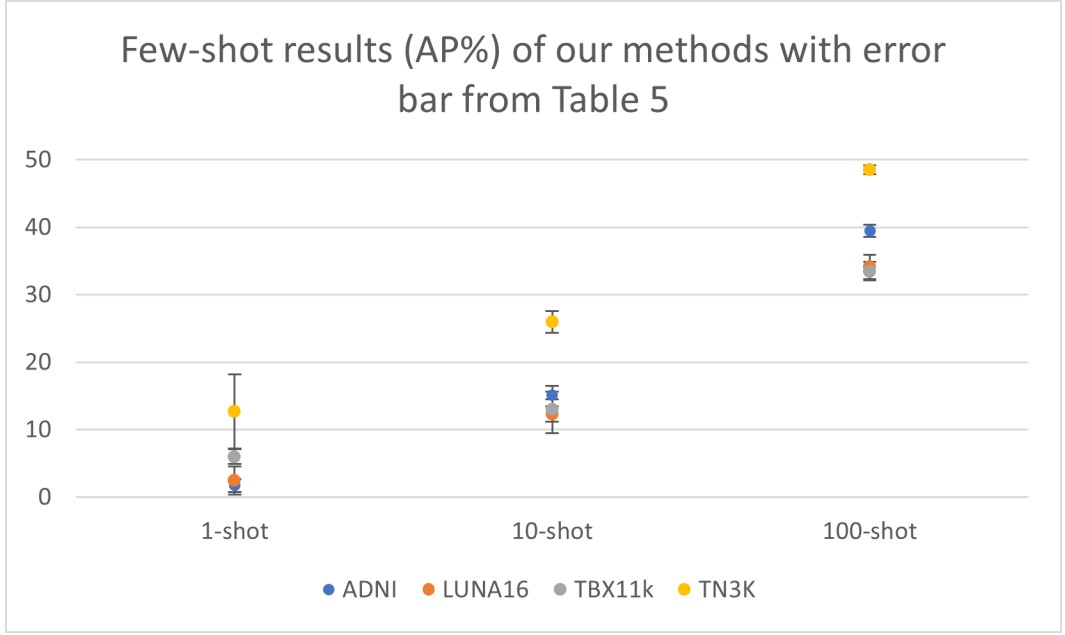

Figure 7: Few-shot results (AP%) of our methods from Table 5 with error bar

## F ZERO-SHOT PERFORMANCE WITH DIFFERENT VLMS

In this section, we present the zero-shot results given various prompts with different VLMs pre-trained on different datasets. As demonstrated, although the zero-shot results are different for these different VLMs, but the pattern of performance increasing with adding expressive attributes still holds. Note that the VLM (O365) is pre-trained with a relative smaller dataset. Furthermore, the GoldG variant is pre-trained on a much larger dataset, which including the dataset VLM (O365) pre-trained on and the extra GoldG dataset.

Table 12: Examples of prompts for different VLMs zero-shot results on CVC-300)

| Pre-trained Data | Attribute | Prompt | AP | AP50 |
|---|---|---|---|---|
| O365 | class | polyp | 2.5 | 3.1 |
| | +shape | polyp irregular shape of bump | 9.2 | 11.3 |
| | +shape +texture +loc | polyp irregular flesh bump in rectum | 9.5 | 12.1 |
| | +shape +texture +loc +modality | colonscope polyp irregular flesh bump in rectum | **13.1** | **16.8** |
| O365+GoldG | class | polyp | 6.1 | 8.3 |
| | +shape | polyp irregular shape of bump | 8.7 | 15.0 |
| | +shape +texture +loc | polyp irregular flesh bump in rectum | 23.4 | 35.6 |
| | +shape +texture +loc +modality | colonscope polyp irregular flesh bump in rectum | **34.9** | **56.3** |

# G  VISUALIZATION

In this section, we provide some visualized examples to illustrate how attribute injection in prompts could affect the object detection for novel objects. In Figure 8, as we include more expressive attributes to the prompts, the predicted bbox can locate the target objects more accurately and confidently.

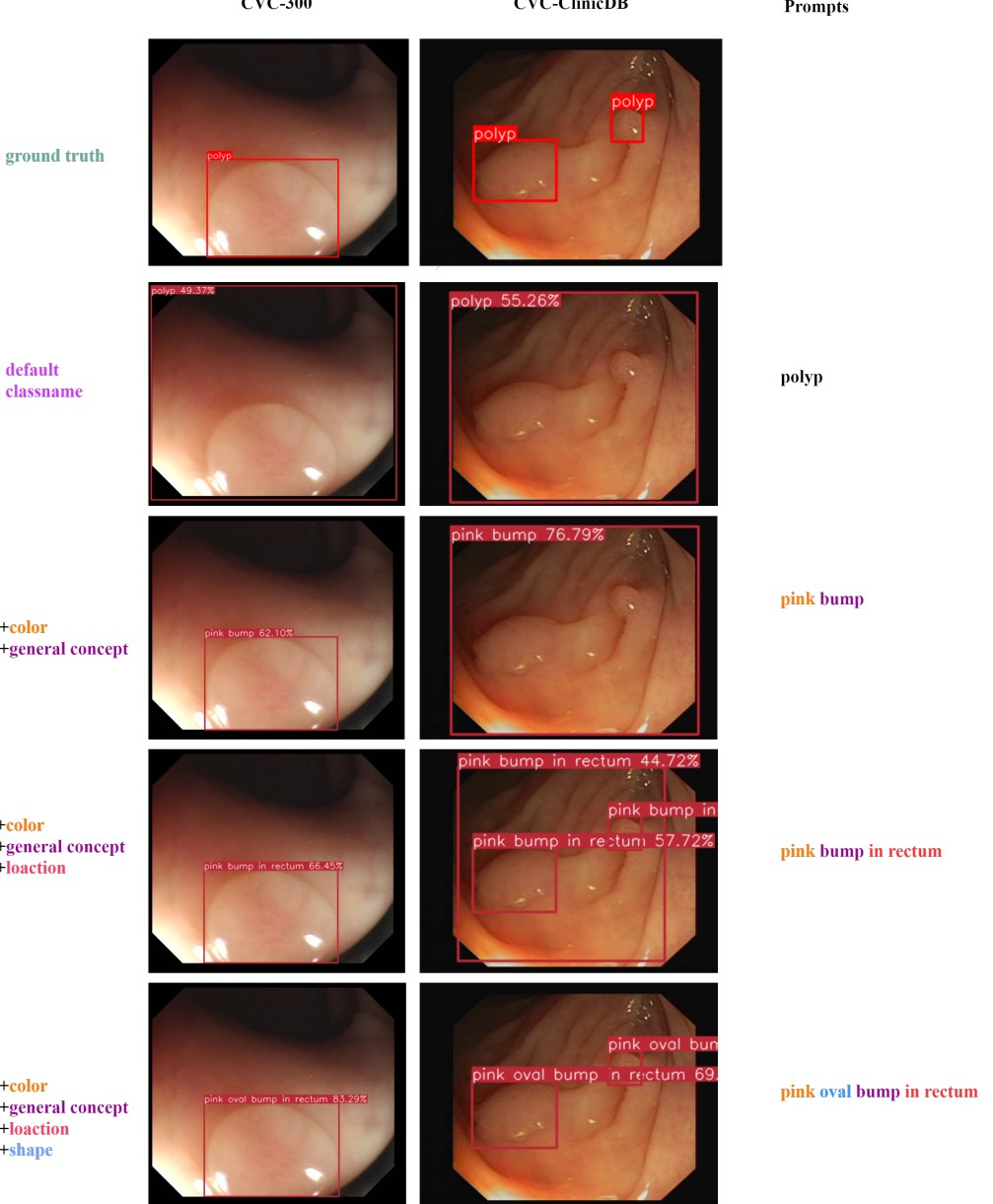

Figure 8: Visualized examples of the effect of including expressive attributes.

As demonstrated in Figure 9, we have shown several images and the predicted bounding boxes under the zero-shot setting on the TN3K dataset. As mentioned before, we directly use the class label as the text prompts for the radiology data, and, in this case, we simply use the 'thryoid nodule' to prompt the pre-trained VLM. As one can see, since the word 'nodule" in the prompt has the language meaning of "small rounded or oval object..." in some context, the predicted bounding box in the zero-shot examples mostly aligned with the salient circle areas in the images. So, these examples prove our presumption that the unseen concept in radiology is too far different from the general image domain, and we need to provide extra visual examples to fine-tune the VLMs.

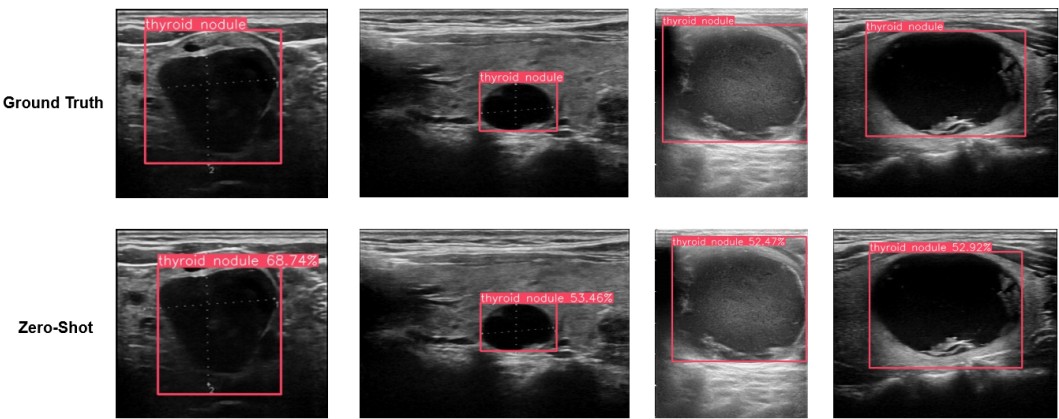

Figure 9: Visualized examples of zero-shot on the radiology dataset.

In Figure 10, we demonstrate a series of images and predicted bounding boxes for 1-shot tasks. As illustrated in the figure, the VLM can quickly understand the pattern of "thyroid nodule", an unseen medical concept, even under a 1-shot setting. We believe the alignment of the visual and language features in the hidden space contributed to such domain-transfer capability of the VLMs. Therefore, even with a single example, the VLM can quickly map the visual features in the example to the given text prompt, and such text prompt can elicit the corresponding visual features during the test time, resulting in a much better performance.

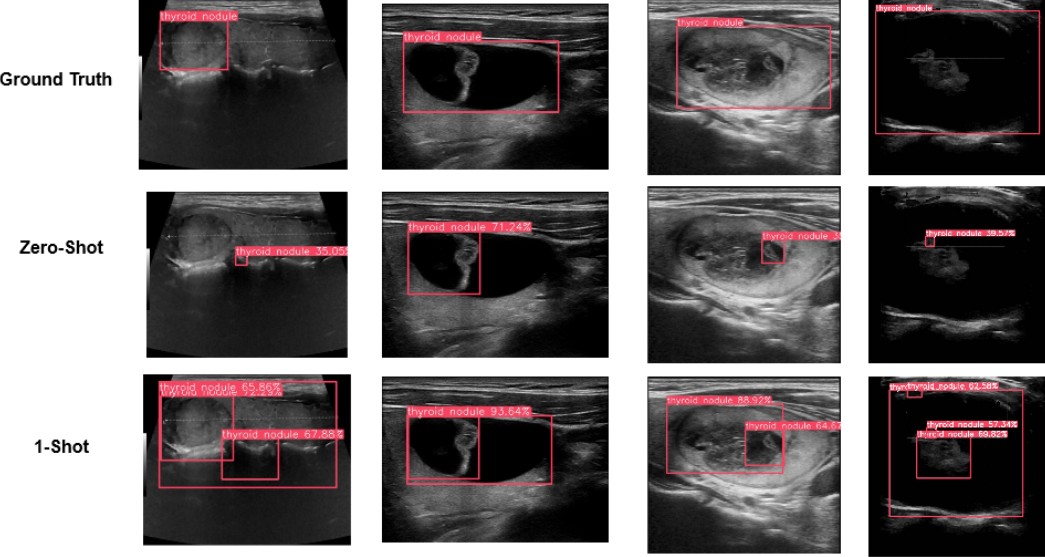

Figure 10: Visualized examples of one-shot results on the radiology dataset.

