# OpenReview forum: "MEDICAL IMAGE UNDERSTANDING WITH PRETRAINED VISION LANGUAGE MODELS: A COMPREHENSIVE STUDY"
_ICLR.cc/2023/Conference — ICLR 2023 poster_

### Official Review · Reviewer_DhmB · 2022-10-24

**Confidence:** 4
**Correctness:** 2
**Technical Novelty And Significance:** 1
**Empirical Novelty And Significance:** 1
**Recommendation:** 6

**Clarity, Quality, Novelty And Reproducibility:**

The paper is well-written indeed. The smooth transition from one section to the next makes the reader feel at ease with the paper. The motivation of the study is quite straightforward. The authors leveraged existing frameworks and provided implementation details. Hence, I believe the study is reproducible. However, in my opinion, the novelty of the study is critically limited.

**Strength And Weaknesses:**

Strength:
- The authors propose several strategies for better elicitation of medical knowledge from vision-language models pretrained on natural images;
- They focus on the design and automatic generation of medical prompts that can include expert-level knowledge and image-specific information;
- Through extensive experiments, the authors support their claims in both zero-shot transfer and fine-tuning scenarios.

Weaknesses:
- There is inadequate justification for explicit requirement of VQA model for deriving attributes/properties;
- Table 5 shows an inconsistent performance disparity between the *-shot and full data.
- Limited technical contribution.


**Summary Of The Paper:**

In this paper, the authors extensively surveyed existing literatures to determine how to leverage the large-scale vision language models effectively for understanding medical images. They claim that a well-designed medical prompt containing domain-specific knowledge is the key to bridging the gap between natural image to medical domains. The authors proposed several approaches to generate medical prompts and demonstrated improved performance using fine-tuned models.

**Summary Of The Review:**

The authors claimed that a well-designed automatic medical prompts generation and leveraging these prompts are the key to elicit knowledge from pre-trained vision language models. However, I am not sure why we need to explicitly derive these attributes using VQA model. While a deep learning model can identify these properties such as shape variance and color difference in an end-to-end setup, I couldn't find a rationale for generating these properties using visual question answering models.

The performance difference between *-shot and full data (Table 5) is debatable. The performance of the proposed model is not very impressive on full data compared to candidate models. In contrast, the performance of *-shot is noticeably better than that of candidate models. The large performance gap appears inconsistent to me.

Overall, in my opinion, the authors make minor modifications to existing approaches for similar tasks. The technical contribution is somewhat limited considering a high bar to ICLR.

---

> ### Author Response · Authors · 2022-11-14
> **Reply to Reviewer DhmB (1/1)**
>
> We thank the reviewer for giving us some appreciation over the extensive experiments/results we provided and some suggestions to improve, but the reviewer's opinion for our contribution generally has some divergence with other reviewers. Our response for the reviewer's opinion is as follows:
>
> ### 1. There is inadequate justification for explicit requirement of VQA model for deriving attributes/properties;
> We thank the review for bringing this question up so we can re-emphasize some merit of VQA models. We introduce VQA model to obtain image-specific attribute information for the unseen concepts, as the language model could only give prior knowledge about the attributes.
>
> ### 2. In the summary, the reviewer said: "The performance difference between *-shot and full data (Table 5) is debatable. The performance of the proposed model is not very impressive on full data compared to candidate models. In contrast, the performance of *-shot is noticeably better than that of candidate models. The large performance gap appears inconsistent to me."
>
> As discussed in our paper, compared to the non-radiology medical images, the domain gap between the radiology and general images is even more significant. We have tried several prompts to do the zero-shot tests for the radiology data. However, the improvement is not as evident as that of the non-radiology data (Table 8 in the updated appendix). So, we presume a handful of visual examples could illustrate the unseen concepts when only providing text prompts is not sufficient to elicit knowledge from the VLMs. Since the language and vision latent space are aligned for VLMs, once a few visual examples provided, the VLM can quickly map the extracted visual features to the given text prompt. Then, when we use such prompt during the inference time, the relevant visual features would be activated to help the VLMs to perform better. In the new version of our paper, Figure 10 in the appendix has proven our point. As
> illustrated in the figure, the VLM can quickly understand the pattern of ”thyroid nodule”, an unseen
> medical concept, even under an 1-shot setting. However, when the number of visual examples is sufficient, models can directly learn all the visual features even without the assistance of text prompts.
>
> ### 3. In the summary, the reviewer thinks our work has limited technical contribution.
> While we can understand reviewers having this thought, but, as Reviewer kq4V said, our work may be incremental from a technical standpoint, but the application and design in the medical domain is new, and our empirical novelty should be appreciated.
> Reviewer Ko8g also mentioned that our work is one of the pioneers in this field. Indeed, our work focuses on validating and showing the development of vision-language techniques can be exploited to tackle the low-resource problem in the medical image domain. While inventing new tools is very important for research, we believe carefully recording and validating the spark made by such tools is also an essential part of the researchers' job and should be appreciated.
>
> ### 4. In the summary, the reviewer also mentioned that "However, I am not sure why we need to explicitly derive these attributes using VQA model. While a deep learning model can identify these properties such as shape variance and color difference in an end-to-end setup, I couldn't find a rationale for generating these properties using visual question answering models."
>
> We thank the reviewer for bringing up this essential question for further discussion and allowing us to provide more explanations. As mentioned before, our work has shown that adding expressive attributes to the text prompts can significantly improve the VLMs' performance in few-shot or zero-shot tasks. However, finding such attributes is not a trivial job, since we need to obtain these attribute values from external knowledge sources. Initially, we either manually design or leverage domain-specific language models to obtain such knowledge, but soon we realized the limitation of such methods. The above approaches could only give us prior knowledge about the attributes of the unseen concepts, however some attribute values may be various in different images. Therefore, we utilize the VQA models to obtain the image-specific attribute values. Though supervised models can extract the color and shape information from the pictures, we need to use a significant amount of data to train them. Contrastively, our approach could inject these information to the text prompts to receive better zero/few-shot performance, as the experiments demonstrate.

---

> > ### Comment · Reviewer_DhmB · 2022-11-18
> > **Acknowledgement on authors' responses**
> >
> > I have read the authors' response as well as other reviews. Most of my concerns seem to have been addressed. Considering clarifications and improvements, I am inclined to accept the paper. I have raised my previous score accordingly.

---

> > > ### Author Response · Authors · 2022-11-18
> > > **Thank you for the response**
> > >
> > > Thank you for the timely reply and the positive feedback. We really appreciate the time and effort spent in this process and feel encouraged by your acknowledgment.

---

### Official Review · Reviewer_kq4V · 2022-10-25

**Confidence:** 4
**Correctness:** 4
**Technical Novelty And Significance:** 2
**Empirical Novelty And Significance:** 3
**Recommendation:** 8

**Clarity, Quality, Novelty And Reproducibility:**

- The paper is well-written.
- Novelty is mostly incremental and related to empirical analysis and application, still appreciated and important.
- Experiments and results and clearly explained and compared to relevant baselines.
- There is no statistical test or multiple runs to form deviations and marginal improvements.
- The nature of the work is incremental from a technical standpoint, however the application and design choices in the medical domain are new.



**Strength And Weaknesses:**

**Strength:**
- The paper is one of the pioneers to explore pretrained VLMs for medical image analysis.
- The application and proposed approach through pretrained transfer learning is relevant and practical.
- The comprehensive study is well-designed.

**Weakness:**
- There are some inconsistencies in the results and occasionally there is no significant improvement vs the baseline (Table 5.)
- Use of some components such as automatic captioning are not well justified.
- Only a single language model has been studied. It would be very interesting to see at least in a limited study how these results are holding for other VLMs.


**Summary Of The Paper:**

The paper ran an empirical study on pretrained vision language models (VLMs) for medical image analysis. They study how to manually design effective medical prompts by using relevant attributes. Results suggest that well-designed prompts can significantly improve the domain transfer capability compared to the default category names.  They evaluated the proposed approach on various existing medical datasets across different modalities including endoscopy, cytology, histopathology and for multiple radiology modality including  X-ray, CT, MRI and Ultrasound. They comprehensively evaluated each modality and dataset for zero-shot, few shot and full fine-tuning  setting. They use the GLIP–T variant as the pretrained VLM and use the PubmedBert-base-uncased variant for language driven prompt generation.

**Summary Of The Review:**

This paper addresses an important application of VLM for medical image analysis and comprehensively studies it. The paper includes comprehensive results and reasonable improvements over baseline.


After rebuttal: as most of my comments has been addressed I increased my score.

---

> ### Author Response · Authors · 2022-11-14
> **Reply to Reviewer kq4V (1/1)**
>
> Before we go into the details of the discussion, we want to say thanks to the reviewer for the positive and constructive comments. In the comments, the reviewer first appreciates our work as one of the pioneers to explore pre-trained VLMs for medical image domain and also thinks our study is well-designed. Another important opinion the reviewer mentioned is: the empirical contribution of the work is still appreciated, especially in medical image domain. For the weakness the reviewer listed in the comment, our replies are as follows:
>
> ### 1. There are some inconsistencies in the results and occasionally there is no significant improvement vs the baseline (Table 5.)
> We thank the reviewer for the pointing out this issue for further discussion. First, we want to emphasize that the main purpose of our work is to demonstrate and validate the powerful domain-transfer capability of VLM models in low-resource and few-shot scenarios. As discussed in our paper, the domain gap between radiology and the natural image domain is relatively large, so we need to provide a few image examples to illustrate the unseen concepts to the VLM. After showing the visual examples, the pre-trained knowledge contained in the VLM could be elicited by the text prompts more efficiently. That's why the VLM performs surprisingly well in the few-shot tests compared to the baseline models. With the increasing of the number of images provided, the supervised learning models will catch up the performance of the VLMs in some cases. Because when the number of provided visual examples is sufficient enough, the domain-transfer capability of VLM might be not that important anymore.
>
> ### 2. Use of some components such as automatic captioning are not well justified.
> We proposed several automatic prompt-generation methods to avoid the tedious and time-consuming process of manually designing prompts. Since we need external knowledge to help us to tell the common color or shape of an unseen concept in the medical domain, we usually spend a lot of time designing one prompt. Therefore, an automatic method is highly desired. In our work, we propose to leverage the large-scale language model such as PubmedBert to obtain external knowledge for each medical concept. We also use VQA models to obtain image-specific information about the object's or concept's expressive attributes. As for the further details of the automatic prompt generation, please refer to the Section B of the updated appendix.
>
> ### 3. Only a single language model has been studied. It would be very interesting to see at least in a limited study how these results are holding for other VLMs.
>
> We thank the reviewer for the constructive suggestion. Following this, we have tried different VLMs pre-trained on various kinds of datasets to see whether the similar results also hold for other VLMs. Though the VLMs trained on different datasets give us different results, but the pattern is the same. We can observe the increase of zero-shot/few-shot performance after we include the expressive attributes in the text prompts even with different VLMs. We have released some extra zero-shot results of the VLMs pre-trained on different datasets in the appendix to show that this pattern holds. Please refer to the section F of the updated appendix for further information. For convenience, we include the table of zero-shot results given various prompts with different VLMs as following:
>
> | Pre-trained data     | Attribute                                                              | Prompt     | AP | AP50 |
> |-----------------------|-----------------------------|-----------------------------|------------------------|--------------------------|
> |        O365        | class          | polyp           | 2.5                    | 3.1                      |
> |        O365             | +shape                                   | polyp irregular shape of bump                   | 9.2                    | 11.3                     |
> |        O365          | +shape +texture +loc             | polyp irregular flesh bump in rectum            | 9.5                    | 12.1                     |
> |         O365         | +shape +texture +loc +modality      | colonscope polyp irregular flesh bump in rectum | **13.1**           | **16.8**          |
> |       O365+GoldG            | class                                                                  | polyp                                           | 6.1                    | 8.3                      |
> |
> |       O365+GoldG        | +shape            | polyp irregular shape of bump                   | 8.7                    | 15.0                     |
> |           O365+GoldG         | +shape +texture +loc            | polyp irregular flesh bump in rectum            | 23.4                   | 35.6                     |
> |          O365+GoldG          | +shape +texture +loc +modality            | colonscope polyp irregular flesh bump in rectum | **34.9**          | **56.3**            |

---

> ### Comment · Reviewer_kq4V · 2022-11-17
> **Acknowledge and Thanks**
>
> Just to acknowledge that I have read your response and have no further questions.
> Thanks for the effort to address my concerns.

---

> > ### Author Response · Authors · 2022-11-18
> > **Thanks for the response**
> >
> > Thank you for the timely reply and the positive feedback. We really appreciate the time and effort spent in this process and feel encouraged by your acknowledgment.

---

### Official Review · Reviewer_Ko8g · 2022-10-25

**Confidence:** 4
**Correctness:** 4
**Technical Novelty And Significance:** 3
**Empirical Novelty And Significance:** 3
**Recommendation:** 8

**Clarity, Quality, Novelty And Reproducibility:**

The paper is reproducible since all datasets and models that are employed were made publicly available, while all hyperparameters are fully described in the paper. It is also of good quality, clear enough and exploring a very hot topic (prompt engineering/injection).

**Strength And Weaknesses:**

*Strengths:

- Extensive experiments using full data, few-shot learning, and zero-shot learning.
- Experiments in a broad range of datasets, varying from photographs to MRIs.
- Well-explained methodology related to prompt generation.

*Weaknesses:

- The authors failed to include images and prompt examples from samples of the radiology datasets.
- The authors do not explain how they use MRI and CT data since they are volumetric images (used as 3D? 2D? Axial, coronal, or sagittal view?).

**Summary Of The Paper:**

The authors use two methodologies to generate textual prompts with information related to the analyzed diseases in order to explore whether the injection of expressive attributes can improve the generalization and performance of the GLIP model. They did experiments using a broad range of medical datasets from various modalities. They showed that the proposed automatic prompt generation methodology improves the generalization and performance in zero and few shot scenarios.

**Summary Of The Review:**

A well-written paper with suitable methodology and novelty. I do not see any significant weaknesses related to the methodology or experiments, but I will point out some things that can improve the paper in a general way:

1) Please show some examples of visualizations and prompts for radiology images, as they are different from the ones already in the paper. It will be an excellent addition (maybe as a part of the supplementary material?).

2) The supplementary material can be improved with more visualization examples and prompts, mainly to show the specificities of each imaging modality.

3) Please add additional details on the experiments with the LUNA and ADNI datasets since MRI and CT are volumetric images.

---

> ### Author Response · Authors · 2022-11-14
> **For reviewer Ko8g (1/2)**
>
> First, we want to thank the reviewer for giving us such positive feedback and appreciation. As the reviewer mentioned, our work explores the VLM's power with extensive experiments on medical datasets across a broad range and modalities, demonstrating the improved generalization and performance in zero and few shot scenarios. We also feel very grateful to receive the reviewer's suggestions to improve our work and our responses are as follows:
>
> ### 1. The authors failed to include images and prompt examples from samples of the radiology datasets. Please show some examples of visualizations and prompts for radiology images, as they are different from the ones already in the paper. It will be an excellent addition (maybe as a part of the supplementary material?)...
>
> We thank the reviewer for the constructive suggestion. Indeed, radiology dataset are very different from non-radiology medical datasets. Following the suggestion, in the updated appendix, we have included more manually designed prompts and the corresponding zero-shot results tested on some of the radiology datasets (Section D of the appendix). We have also provided more visualization examples for the radiology data in the updated appendix (Section G).
>
> **[prompt for radiology data]**
> As the reviewer asked, we provide a table (Table 8 in the updated appendix) of the zero-shot results with different manually designed prompts for radiology dataset. It's quite apparent to see that the deliberately designed prompts still improve the performance, but the enhancement is not as strong as that of the non-radiology datasets. Furthermore, the attributes we used for radiology concepts are not the same as those we used for non-radiology concepts, since some attributes, such as color, do not exist in this modality. Generally speaking, we have tried various prompts for radiology datasets, but compared to the default class label, the improvement is not as significant as that of the non-radiology datasets. So, we decide to use the default class label as prompts to finetune the VLM (Vision-Language Models) on the radiology datasets.
>
> |                                            | Prompt                                      | AP | AP50 |
> |---------------------------------------------|------------------------------------------------------------------------|------------------------|--------------------------|
> |                                      | thyroid nodule                                                         | 1.9                    | 4.2                      |
> | wikipedia     | thyroid nodules  are nodule which commonly arise within   |5.6 | 10.8
> | + description | **irregular** thyroid tumor.                                      | 4.8                    | 11.2                     |
> |                + description                             | **salient** thyroid tumor.                                        | 5.5                    | 11.1                     |
> | + domain                | thyroid tumor in **medical imaging.**                             | 11.2                   | 20.3                     |
> |            + domain                                  | thyroid tumor in **medical ultrasound**  imaging.                 | 11.3                   | 20.9                     |
> | + description + domain | **salient** thyroid tumor in **medical ultrasound  imaging** |          **12.2**        | **21.4**            |
>
> **[visualization analysis]** In the updated version of our paper, we have included a few more visualization examples of the VLMs performance on radiology datasets in the appendix. We also added some analysis for the visualization examples in the appendix, and we will briefly summarize our analysis in the comment.
>
> In Figure 9, we have shown several image examples from the T3NI dataset and the predicted bounding boxes under the zero-shot setting. As mentioned before, we directly use the class label as the text prompts for the radiology data, and, in this case, we simply use the default class label, `thyroid nodule', to prompt the pre-trained VLM. As one can see, since the word "nodule" in the prompt has the language meaning of "small rounded or oval object..." in some context, the predicted bounding box in the zero-shot examples mostly aligned with the salient round areas in the images. However, a shape of thyroid nodule is varying from image to image. So, these examples prove our presumption that the unseen concept in radiology is too different from the general image domain, and we need to provide extra visual examples to fine-tune the VLMs. Therefore, in Figure 10, we demonstrate a series of images and predicted bounding boxes for zero and one-shot tasks. As illustrated in the figure, the VLM can quickly understand the pattern of "thyroid nodule", an unseen medical concept, even under a 1-shot setting. As we analysis in our paper, the reason for such strong performance are mainly attributed to the amazing domain-transfer capability of the VLMs.

---

> > ### Author Response · Authors · 2022-11-14
> > **For reviewer Ko8g (2/2)**
> >
> > ### 2.The authors do not explain how they use MRI and CT data since they are volumetric images (used as 3D? 2D? Axial, coronal, or sagittal view?)" and the suggestion 3 in the summary part:
> > We thank the reviewer for the detailed suggestion on the missing description of the dataset. Since the pre-trained model only takes 2D input, for convenience, we convert the 3D volumetric data to 2D images by slicing the data from the axial view.

---

> > > ### Comment · Reviewer_Ko8g · 2022-11-29
> > > **Response to authors**
> > >
> > > Dear authors, I am fully satisfied with your response to my questions, doubts, and suggestions. Thank you.

---

### Official Review · Reviewer_r5bj · 2022-10-31

**Confidence:** 4
**Correctness:** 3
**Technical Novelty And Significance:** 2
**Empirical Novelty And Significance:** 3
**Recommendation:** 6

**Clarity, Quality, Novelty And Reproducibility:**

Clarity:

Typos and nits below
> moves toward the ear of large-scale pre-trained models
Should 'ear' be era instead?

> mismatch between domains may compromise the capability of the pre-trained models being transferred from one to another
Consider adding Raghu et al 2019 citation - https://proceedings.neurips.cc/paper/2019/file/eb1e78328c46506b46a4ac4a1e378b91-Paper.pdf here although it is discussed further in related work.

In Related work, consider also adding https://arxiv.org/abs/2105.11333

>  However, our preliminary results obtained from the VQA prompts suggest that certain attribute (e.g., location) may not be appropriated
answered by the pre-trained VQA models
appropriated -> appropriately


Reporting the average performance across datasets in Table 2 could be helpful

Quality

The paper is well written and easy to follow for the most part.


Novelty
The idea of using VLMs trained on natural images and augmented with domain specific LMs and VQA models seems interesting and novel to me.

Reproducibility

> We freeze the bottom two layers of the image encoder and decay the learning rate by 0.1 when the validation performance plateaus.

Why was this chosen and what is the result of this?


The backbones differ between the baselines (RN50) and the proposed methods (SwinT). This is an additional confounding factor.


I am also a bit surprised by the margin of improvement obtained by finetuned model using a single example in Table 5. Some explanation here would be helpful.

I do not see code details included and the appendix is quite sparse.

My major question is regarding the results in Table 2. Does this involve any finetuning with the generated prompts? If not, the zero shot results seem surprisingly strong. It may be good to add further explanations as to why this is the case.


**Strength And Weaknesses:**

- Strengths:
1. The paper proposes an interesting approach to adapt pre-trained VLMs to the medical domain using prompts generated by domain specific LMs and VQA models.
2. The results across 13 datasets are quite extensive and suggest strong improvements.
3. The writing is easy to follow for the most part

- Weaknesses.
1. No error bars in the results so hard to ascertain statistical significance.
2. No code reproducibility and details on the prompts generation are quite sparse. Similarly some results seem surprisingly strong and could do with additional explanation.

**Summary Of The Paper:**

This paper studies the applicability of pre-trained vision language models to the medical domain. The paper shows how manually designed expressive prompts can bridge the domain gap between natural and medical images. Further, they propose methods for automatic prompt generation and demonstrate across a range of medical datasets for the lesion detection task that these prompts are effective. As a final result, the authors also demonstrate that their fine-tuned models are better than the supervised counterparts.

**Summary Of The Review:**

The paper proposes an interesting approach to adapting pretrained VLMs to the medical domain using prompts generated by domain specific LMs and VQA models. Results across 13 medical datasets suggest the potential of the method.

Overall the approach is interesting although not particularly novel. The authors should consider adding more explanations and analysis (especially qualitative) to demonstrate why their proposed method is so effective especially Table 2 results in the absence of finetuning.

Further, error bars are missing in results. Although 13 datasets are considered, they all contain a single task and it might interesting to consider classification or medical VQA tasks too for the paper.

Overall, if the authors can satisfactorily respond to the questions above, willing to reconsider my score.

---

> ### Author Response · Authors · 2022-11-14
> **Reply to Reviewer r5bj (1/2)**
>
> ## For Reviewer r5bj
> Before we start discussing the detail of the comments, please allow us to thank the reviewer for making time to give us such insightful and helpful feedback so we can improve our work even better. We are glad to see that the reviewer thinks our work is interesting in approach and appreciates the extensive experiments we conducted. In the following, we respond to the comments point by point.
>
> ### 1. No error bars in the results so hard to ascertain statistical significance.
> We thank the reviewer for the constructive suggestion. Following the suggestion, we have included the error bars for most of the results that appeared in Tables 2, 3, and 5 (please refer to the section E of the updated appendix due to the space limit). For convenience, we have also given an example table in the following for the few-shot learning results where we tested several trials with different random seeds for our models during experiments.
>
> |   | Method | CVC-300         | CVC-ClinicDB    | CVC-ColonDB     | Kvasir          | ETIS            | Avg   |
> |---|-------------------------|-----------------|-----------------|-----------------|-----------------|-----------------|---------------|
> |   | GLIP-T                  | 69.6 $\pm$2.42 | 59.4$\pm$1.63 | 52.3$\pm$0.38 | 63.0$\pm$1.44 | 43.6$\pm$3.69 | 57.6 |
> |   | Ours (Manual)           | 70.2$\pm$1.96 | 61.6$\pm$0.88 | 53.6$\pm$2.61 | 66.8$\pm$2.63 | 51.8$\pm$1.94 | **60.8** |
> |   | Ours (Auto)             | 71.3$\pm$0.93 | 60.4$\pm$1.25 | 55.4$\pm$2.36 | 67.1$\pm$1.31 | 49.6$\pm$4.31 | **60.8** |
>
> ### 2. No code reproducibility and details on the prompts generation are quite sparse.
> We will release our code for reproducibility upon the acceptance of our submission. For more details on the prompt generation, we have also included a new section on the prompt generation details (Section B of the appendix) and more visualized examples (will explain more in the next few responses) in the updated appendix.
>
> ### 3. Typos and reference update.
> Thanks for carefully reading our paper and helping us to improve. We have modified those typos and minor errors in the updated version. For the reference, we have also added the MedViLL paper (https://arxiv.org/abs/2105.11333) in the related work, since we also think this research work aligned with our topic and is a pioneer work introducing a vision-language approach to the medical image domain as well.
>
> ### 4. Reporting the average performance across datasets in Table 2 could be helpful.
> Thanks for the suggestion. For Table 2, we have included the column of average values in the updated version.
>
> ### 5. Questions on the layer freeze and learning rate decay.
> We thank the reviewer for the detailed question on the implementation. As we mentioned in the ablation studies section, freezing the bottom two layers (of the vision encoders) during finetuning is a similar setting also suggested in the GLIP work. We also find in our experiments that this setting gives us better results. And we also follow their setting in the learning rate decay which is a common trick used in many deep learning algorithms during training.
>
> ### 6. The backbones differ between the baselines (RN50) and the proposed methods (SwinT). This is an additional confounding factor.
> We totally agree with the reviewer on the disparity between different backbones, and therefore, we have already included the comparisons between the supervised method and our approach that both use the Swin-T as the backbone, \ie, the DyHead model in Table 2, 3, and 5 for supervised method is with the Swin-T backbone. The results indicate that our proposed method also surpasses the baseline with the SwinT backbone.

---

> > ### Author Response · Authors · 2022-11-14
> > **Reply to Reviewer r5bj (2/2)**
> >
> > ### 7. More explanations on one-shot learning: ``I am also a bit surprised by the margin of improvement obtained by finetuned model using a single example in Table 5. Some explanation here would be helpful.''
> >
> > We thank the reviewer for bringing this up for further discussion and allowing us to provide more explanations. Table 5 presents the fine-tuning results on the radiology datasets. First, as mentioned in our paper, we think that the domain gap between the radiology and natural image domain is relatively large, even larger than the gap between non-radiology medical and natural image domains. We tried several prompts on the radiology datasets to illustrate the unseen concepts for zero-shot tests, however, the performance is not as good as that of the non-radiology medical datasets (check Table 8 in the updated appendix). Inspired by some recent research works that suggest that image could also be treated as language input [1], [2], we provided at least one image samples to illustrate this unseen concept. As the experiments demonstrate, the VLM (Vision-Language Models) can learn fast and give a strong result even with one example. The reason for this observation is also what our work dedicate to discovering and validate. As discussed in the paper, we believe the alignment of the visual and language features in the hidden space contributed to such domain-transfer capability of the VLMs. Thus, even with a single example, the VLM can quickly map the visual features in the example to the given text prompt, and such text prompt can elicit the corresponding visual features during the test time, resulting in a much better performance.
> >
> > To show the transfer capability of the VLMs, we have also included the visualized examples of the comparison of zero-shot and one-shot tests on radiology dataset. As demonstrated in Figure 10 in the updated appendix, even the zero-shot result can not detect the target area, the one-shot result partially find the object, with only one image example provided. This example demonstrates that the one-shot fine-tuned VLM can quickly learn the important features with the help of the text prompt.
> >
> > ### 8. More explanations on the surprisingly strong zero-shot performance: ``My major question is regarding the results in Table 2. Does this involve any finetuning with the generated prompts? If not, the zero shot results seem surprisingly strong. It may be good to add further explanations as to why this is the case."
> >
> > We thank the reviewer for pointing out this essential question for further discussion and allowing us to provide more explanations.
> >
> > First, we need to make clear that our data was not leaked or exposed to the model in any form before the zero-shot test, which means we didn't fine-tune the VLMs before the zero-shot tests. And we only fine-tuning the VLMs with the best generated prompts during few-shot learning. And for the text-generation model such as PubmedBERT or OFA models, we also only use them as the out-of-box settings.
> >
> > We were also surprised by the zero-shot performance of our approach on some datasets, for example, the CVC-300 dataset. Therefore, we demonstrated the visualized process of adding expressive attributes to the prompts and investigated how this would benefit the model's performance. As demonstrated in Section D of the updated appendix, the bounding boxes given by the same VLM model narrowed down from covering the whole picture to very close to the ground truth just by adding few attributes, such as "pink" or "bump", to the text prompt. This visualized process again proved our intuition that the expressive visual attribute representation is almost invariant in different domains and can help domain-transfer tasks.
> >
> > ### 9. Future work suggestion on including more tasks such as classification or medical VQA tasks.
> > We thank the reviewer for pointing out the future direction.
> > As one of the pioneering works which apply the vision-language approaches in the medical image domain, we choose to present the power of text prompts and VLMs by first focusing on the object detection tasks in this work. Therefore, we provide extensive experiments across different datasets to support our discovery. In fact, we will continue to explore this direction using other VLMs, such as CLIP, to tackle the classification problem in the medical image domain. Due to the length limitation, we leave other different visual tasks to our future work.
> >
> > ## Reference
> > - [1] [Wenhui Wang, Hangbo Bao, Li Dong, Johan Bjorck, Zhiliang Peng, Qiang Liu, Kriti Aggarwal, Owais Khan, Saksham Singhal, Subhojit Som, & Furu Wei (2022). Image as a Foreign Language: BEIT Pretraining for All Vision and Vision-Language Tasks](https://arxiv.org/abs/2208.10442)
> > - [2] [Rinon Gal, Yuval Alaluf, Yuval Atzmon, Or Patashnik, Amit H Bermano, Gal Chechik, & Daniel Cohen-Or (2022). An Image is Worth One Word: Personalizing Text-to-Image Generation using Textual Inversion](https://arxiv.org/abs/2208.01618)

---

> > > ### Comment · Reviewer_r5bj · 2022-11-18
> > > **thank you for the detailed rebuttal**
> > >
> > > I would like to thank the authors for their hard work and detailed responses to my questions. I do not have further questions at this point and have updated my scores.

---

> > > > ### Author Response · Authors · 2022-11-18
> > > > **Thanks for the response**
> > > >
> > > > Thank you for the timely reply and the positive feedback. We really appreciate the time and effort spent in this process and feel encouraged by your acknowledgment.

---

### Author Response · Authors · 2022-11-15
**General Response**

We appreciate all the reviewers for their timely and detailed reviews and constructive comments. These comments drive us to improve our work. We have noticed that most of the reviewers think our approach is interesting (R1, R2, R3) and is one of the few works focusing on applying vision-language methods to the medical domain (R2, R3). All reviewers (R1, R2, R3, R4) found our paper well-written and easy to follow, and the extensive experiments across multiple medical domains convince the reviewers that our claim is well-supported. We have replied to the reviewers for their concerns and doubt individually, and we are happy to address any further questions during the discussion period.

In the updated version of our paper for the rebuttal phase, we have corrected the typos and minor errors (Thank R1 for the detailed review) and included more implementation detail, error bars for experiment results (As asked by R1), results for different VLMs (As asked by R3), and visualization examples for radiology data (As asked by R2) in the appendix. To address the few-shot and full-data performance problem in Table 3 raised by both R3 and R4, we provide our detailed explanation in the individual response and more visualized results in the appendix.

---

### Comment · Area_Chair_feyC · 2022-11-29
**Please acknowledge and respond to authors responses**

Dear reviewers,

Please make sure that you review the author responses and acknowledge that you have read them (and respond and make any adjustments to your rating as suitable) ASAP, if you have not already done so. This is an important part of the process and authors have put a lot of time to prepare responses. We also need to make decisions based on the final reviewer assessments very soon.

In this year’s reviewing process, a few borderline papers will also be selected to have a virtual meeting with reviewers to discuss. Your responses to the author rebuttal will help to determine which papers need (or do not need) this discussion.

Thank you,

Paper AC

---

### Decision · Program_Chairs · 2023-01-20

**Decision:**

Accept: poster

**Justification For Why Not Higher Score:**

This is a solid paper but the technical novelty is still somewhat limited.

**Justification For Why Not Lower Score:**

I do not recommend reject, since this paper contributes significant insights into the application of pre-trained vision language models to the medical domain that will be of interest to many researchers. It was also given strong reviews by all four reviewers.

**Metareview: Summary, Strengths And Weaknesses:**

All reviewers concurred that this paper is above the bar for acceptance. Reviewers particularly noted the value of the work as a strong and extensive study of applying pre-trained vision language models to the medical domain. They also found the approach for domain specific generation of prompts to be interesting, and the paper to be well written and clear. Although some reviewers noted that the technical novelty was somewhat limited, they still considered the paper a significant enough contribution for acceptance as an early work in studying in-depth the application of pre-trained vision language models to medical images. I agree with the reviewer assessments and recommend acceptance for this paper.

**Note From Pc:**

if the above contains the word "oral" or "spotlight" please see: "oral" presentation means -> notable-top-5% and "spotlight" means -> notable-top-25%. As stated in our emails, we are disassociating presentation type from AC recommendations